# A Novel Coupling Model of Physiological Degradation and Emotional State for Prediction of Alzheimer’s Disease Progression

**DOI:** 10.3390/brainsci12091132

**Published:** 2022-08-25

**Authors:** Jiawei Yang, Shaoping Wang

**Affiliations:** 1School of Automation Science and Electrical Engineering, Beihang University, Beijing 100191, China; 2Beijing Advanced Innovation Center for Big Data-Based Precision Medicine, Beihang University, Beijing 100191, China

**Keywords:** Alzheimer’s disease, prediction method, disease progression model, stochastic process

## Abstract

The prediction of Alzheimer’s disease (AD) progression plays a very important role in the early intervention of patients and the improvement of life quality. Cognitive scales are commonly used to assess the patient’s status. However, due to the complicated pathogenesis of AD and the individual differences in AD, the prediction of AD progression is challenging. This paper proposes a novel coupling model (P-E model) that takes into account the processes of physiological degradation and emotional state transition of AD patients. We conduct experiments on synthetic data to validate the effectiveness of the proposed P-E model. Next, we conduct experiments on 134 subjects with more than 10 follow-ups from the Alzheimer’s Disease Neuroimaging Initiative. The prediction performance of the P-E model is significantly better than other state-of-the-art methods, which achieves the mean squared error of 7.137 ± 0.035. The experimental results show that the P-E model can well characterize the non-monotonic properties of AD cognitive data and can also have a good predictive ability for time series data with individual differences.

## 1. Introduction

Alzheimer’s disease (AD) is the most common type of dementia and a growing health problem. AD causes the brain to atrophy and brain cells to die, resulting in a continuous decline in thinking, behavioral, and social skills, which affects a person’s ability to function independently. Proactive management of AD can improve the quality of life of affected individuals and their caregivers [1]. Therefore, for effective disease management, it is crucial to accurately assess and predict AD progression. In recent studies, various clinical data were used to analyze AD progression, containing imaging [2,3,4], generic [5,6], clinical, and cognitive data [7,8]. Due to the complexity of AD pathogenesis, the challenge lies in delineating the relationships among various types of clinical data. Therefore, more suitable models are required to characterize AD progression and help researchers further understand AD pathogenesis.

Generally, the development of Alzheimer’s disease is divided into three stages: (1) cognitive normal (CN), (2) mild cognitive impairment (MCI), and (3) Alzheimer’s disease (AD). With the development of Alzheimer’s disease, a subject’s physiological indicators and cognitive scales will change in a certain trend. Therefore, recent models of AD progression can be divided roughly into two categories: (a) those that directly describe the stages of Alzheimer’s disease [9] and (b) those that consider the stages of Alzheimer’s disease as latent variables and the physiological indicators or cognitive scales as observed variables [10,11]. These two categories of models usually assume the developmental stages of AD to be discrete states and use Markov chains to characterize the transition between different stages. However, there is no uniform setting for the number of states, which is usually determined by model selection. The resulting problems include greatly increased computational costs and missing interpretability.

Emotion processing dysfunction is associated with the cognitive decline and physiological deterioration in AD. Johansson et al. [12] used regression analyses to show that apathy and anxiety are associated with cognitive decline as early clinical manifestations of AD. Arroyo-Anlló et al. [13] found significantly lower total results for cognitive and emotional assessments in the AD group than those in the CN group via an investigation of music processing. Chaudhary et al. [14] reviewed studies related to emotion processing dysfunction in Alzheimer’s disease. Their review study highlighted the pronounced effect of positive emotion on enhancing the memory and emotional processing deficits related to biomarkers of hippocampal circuit dysfunction. In summary, it shows that emotion processing is affected by AD, and positive emotions can also improve AD symptoms. However, to the best of our knowledge, there are still no studies that take emotional changes into account in the models for AD prediction.

In the task of AD prediction, the challenge is to build a model that not only can achieve high-performance AD prediction, but also can help researchers further understand AD pathogenesis. At present, many studies consider the development of AD to be irreversible [15,16,17], but it usually does not show a monotonic decline in cognitive data. Excluding the test error of the subjects, we believe that emotional fluctuations can cause both improvement and deterioration in cognitive test results. Moreover, there is a coupling relationship between the emotional state transition and physiological degradation processes. Therefore, we propose a novel coupling model of physiological degradation and emotional state (P-E model) for AD prediction. In the P-E model, cognitive data are observations that can be obtained directly, while the physiological degradation process and emotional state transition process are continuous, hidden states. Physiological degradation is characterized by a monotonic stochastic process, thus exhibiting irreversible properties of development in AD. The emotional state transition is characterized by a non-monotonic stochastic process. The observation process is jointly determined by the physiological degradation process and the emotional state transition process. It can describe the non-monotonic degradation process and is more suitable for the real observation results.

## 2. Related Work

Longitudinal data track the same type of information on the same subjects at multiple points in time, which can provide temporal features to disease progression models (DPMs). Mofrad et al. [18] utilized mixed-effects models to derive the features of trajectory changes from longitudinal data containing the Rey Auditory Verbal Learning Test (RAVLT) and the Alzheimer’s Disease Assessment Scale-Cognitive Subscale (ADAS-cog13). The derived features were fed into classification models to predict the conversions from CN to MCI or from MCI to AD. Lorenzi et al. [19] proposed a DPM based on Bayesian Gaussian process regression, which is capable of probabilistic diagnosis and estimation of the biomarker trajectories with cognitive data and magnetic resonance imaging (MRI) measurements. Zhang et al. [20] used a three-state Markov model to model the progression of AD. The model predicted transitions between different clinical diagnosis states with risk factors such as age, apolipoprotein E4 (APOE4), and brain volumetric MRI features, whereas Kang et al. [21] developed a Bayesian hidden Markov model, which uses clinical diagnosis states as unobservable latent variables and functional assessment questionnaire (FAQ) scores as observable variables. The hidden Markov model identified four states of AD pathology, which were determined by deviance information criteria (DIC) [22], and examined the effects of the hippocampus, age, gender, and APOE4 on the degradation of cognitive function.

Recently, deep learning is gaining a lot of popularity because it can solve an extremely wide range of problems in several domains with good accuracy. Without too much professional knowledge, deep learning can automatically implement the process of feature extraction from data, rather than engineering features manually. Jung et al. [23] proposed a deep recurrent network that utilizes cognitive data from the Mini-Mental State Examination (MMSE), ADAS-cog11, and ADAS-cog13 as the inputs. Along with the MRI features, the model predicted individuals’ cognitive scores and clinical labels over time. Mehdipour Ghazi et al. [24] proposed a training algorithm for long short-term memory (LSTM) networks to improve their robustness against missing data. An LSTM network trained by the proposed algorithm was utilized to predict AD progression using the longitudinal measurements of six volumetric MRI biomarkers. Lee et al. [25] used a gated recurrent unit (GRU) to suggest that the model with longitudinal data may achieve better classification results than that with baseline data in the prediction of MCI-to-AD conversion.

From the aforementioned works, we can observe that there is a problem of determining the number of discrete states to describe the state transition of AD. On the other hand, the development of AD is not a step but a continuous process. Although deep learning methods can reduce the need for professional knowledge, they also reduce the interpretability of the model. In order to further understand AD pathogenesis, we need more physiological meanings for the changes described by the model. In this paper, we propose the P-E model, using a physiological degradation process and an emotional state transition process to jointly describe the underlying AD development. Cognitive data are used as observations to assess and predict the progression of AD. The main contributions of this paper are as follows: (1) We develop a model based on gamma process to characterize the monotonic physiological degradation process underlying AD progression. (2) We establish the relationship between emotional changes and physiological degradation in AD progression, and we correlate cognitive data with the emotional state transition and physiological degradation processes. (3) We carry out experiments to demonstrate the validity of the P-E model and discuss the effect of the parameters on the model. Moreover, we verify its applicability with a real AD dataset.

## 3. Methodology

In this paper, two basic hypotheses are proposed: (1) The physiological degradation process in AD patients is irreversible, and (2) there is a strong coupling relationship between the patient’s emotional state and the physiological degradation process. As shown in Figure 1A, the physiological degradation process D(t) is characterized by a monotonically increasing process, zero means that the physiological degradation does not occur, and the degradation rate is affected by the emotional state. As shown in Figure 1B, the emotional state transition process M(t) may increase or decrease, influenced by the physiological degradation stage and positive or negative emotions. In this way, there is a strong coupling relationship between the physiological degradation process and the emotional state transition process.

The physiological degradation process is a monotonic stochastic process denoted by {D(t),t≥0}, where D(t)≥0. D(t)=0 indicates that physiological degradation has not occurred, and as the D(t) increases, the physiological degradation becomes more serious. The emotional state transition process is a non-monotonic stochastic process {M(t),t≥0}, where M(t)∈R. Observation time points are known as $t_{0},t_{1},\ldots ,t_{n}$. The physiological degradation stage $D(t_{i})$ is related to the physiological degradation stage $D(t_{i-1})$ and emotional state $M(t_{i-1})$ of the previous observation time point. The emotional state $M(t_{i})$ is related to the emotional state $M(t_{i-1})$ of the previous observation time point, the physiological degradation stage $D(t_{i-1})$, and the emotional impact $\delta \left(t_{i}\right)={\left[\delta^{\left(-\right)}\left(t_{i}\right)\;\delta^{\left(+\right)}\left(t_{i}\right)\right]}^{T}$. In this paper, the emotional impacts are considered to be divided into negative emotions $\delta^{\left(-\right)}\left(t_{i}\right)$ that help improve symptoms and positive emotions $\delta^{\left(+\right)}\left(t_{i}\right)$ that accelerate the deterioration of symptoms. The observation process is denoted by {Y(t),t≥0} and is related to both the physiological degradation process and the emotional state transition process. A graphical representation of the whole model is shown in Figure 2.

### 3.1. Emotional State Transition Process

The emotional state transition process is affected by emotional impacts and physiological degradation. In this paper, it is assumed that emotional impacts occur randomly and are cumulative. The $l^{\left(-\right)}{}^{th}$ negative emotion is denoted by $\delta_{l^{\left(-\right)}}^{\left(-\right)}\left(t\right)$, and the counting process $\left\{N^{\left(-\right)}\left(t\right),t\geq 0\right\}$ follows the Poisson process with the parameter $k^{\left(-\right)}$. The impact intensity follows the exponential distribution with the parameter $\lambda^{\left(-\right)}$. Similarly, the $l^{\left(+\right)}{}^{th}$ positive emotion is denoted by $\delta_{l^{\left(+\right)}}^{\left(+\right)}\left(t\right)$, and the counting process $\left\{N^{\left(+\right)}\left(t\right),t\geq 0\right\}$ follows the Poisson process with the parameter $k^{\left(+\right)}$. The impact intensity follows the exponential distribution with the parameter $\lambda^{\left(+\right)}$. During a given time period $\left[t_{i-1},t_{i}\right)$, the cumulative negative emotion is $\delta^{\left(-\right)}\left(t_{i}\right)=\sum_{l=N^{\left(-\right)}\left(t_{i-1}\right)}^{N^{\left(-\right)}\left(t_{i}\right)}\delta_{l}^{\left(-\right)}\left(t\right)$, the cumulative positive emotion is $\delta^{\left(+\right)}\left(t_{i}\right)=\sum_{l=N^{\left(+\right)}\left(t_{i-1}\right)}^{N^{\left(+\right)}\left(t_{i}\right)}\delta_{l}^{\left(+\right)}\left(t\right)$, and the emotional impact over time period $\left[t_{i-1},t_{i}\right)$ is $\delta \left(t_{i}\right)={\left[\delta^{\left(-\right)}\left(t_{i}\right)\;\delta^{\left(+\right)}\left(t_{i}\right)\right]}^{T}$. The counting process is formulated by
(1)$$f_{\Delta N^{\left(\cdot \right)}\left(t_{i}\right)}\left(n_{i}|k^{\left(\cdot \right)}\right)=k^{\left(\cdot \right)}\Delta t_{i}\exp\left\{-n_{i}k^{\left(\cdot \right)}\Delta t_{i}\right\},$$
where $\Delta N^{\left(\cdot \right)}\left(t_{i}\right)=N^{\left(\cdot \right)}\left(t_{i}\right)-N^{\left(\cdot \right)}\left(t_{i-1}\right)$, and $\Delta t_{i}=t_{i}-t_{i-1}$. The impact intensity is formulated by
(2)$$f_{\delta^{\left(\cdot \right)}}\left(\delta |\lambda^{\left(\cdot \right)}\right)=\lambda^{\left(\cdot \right)}\exp\left\{-\lambda^{\left(\cdot \right)}\delta \right\}.$$

The emotional state at time *t* is represented by a continuous variable M(t), which can change for better or worse over time. In order to be consistent with the direction of physiological degradation, we define that an increase in the emotional state exacerbates the process of physiological degradation, and a decrease in the emotional state moderates the process of physiological degradation. The Wiener process is a commonly used non-monotonic stochastic process, which is suitable to describe the transition of emotional state. This paper proposes an emotional state transition process based on the Wiener process. From time $t_{i-1}$ to time $t_{i}$, the change in emotional state is formulated as follows: (3)$$M\left(t_{i}\right)=M\left(t_{i-1}\right)+\mu_{M}\left(t_{i},t_{i-1}\right)+\sigma B\left(\Delta t_{i}\right),$$
where $B(\Delta t_{i})$ is a standard Brown motion, and $\mu_{M}\left(t_{i},t_{i-1}\right)$ is the drift term defined as: (4)$$\mu_{M}\left(t_{i},t_{i-1}\right)=c_{\mu }\tanh\left(\omega_{\delta }^{T}\delta \left(t_{i}\right)+\omega_{z}^{T}z\left(t_{i-1}\right)+\omega_{d}D\left(t_{i-1}\right)+\omega_{0}\right)\Delta t_{i}$$
where $\mu_{M}\left(\cdot \right)$ is a function of the physiological degradation stage $D(t_{i-1})$ at the last observation time, the emotional impact $\delta (t_{i})$, and the covariate vector $z(t_{i-1})$. Here, $z(t_{i-1})$ contains covariates that affect the emotional state transition process, which characterizes individual differences between subjects; $\omega_{\delta }$, $\omega_{z}$, $\omega_{d}$ and $\omega_{0}$ are regression coefficients; and tanh(·) is one of the commonly used activation functions, belonging to the S-like functions that suppress the input values to a bounded range. Then, $c_{\mu }\tanh\left(\cdot \right)$ is the drift coefficient capturing the rate of transition, bounded between $-c_{\mu }$ and $c_{\mu }$. Thus, given that $\delta (t_{i})$, $z(t_{i-1})$ and $D(t_{i-1})$, the probability density function of $\Delta M\left(t_{i}\right)=M\left(t_{i}\right)-M\left(t_{i-1}\right)$ is formulated as follows: (5)$$\begin{array}{cc} \; & f_{\Delta M(t_{i})}\left(\Delta m_{i}|\delta \left(t_{i}\right),z\left(t_{i-1}\right),D\left(t_{i-1}\right)\right)\\ \; & =\frac{1}{\sqrt{2\pi \Delta t_{i}}\sigma }\exp\left\{-\frac{{\left(\Delta m_{i}-c_{\mu }\tanh\left(\omega_{\delta }^{T}\delta \left(t_{i}\right)+\omega_{z}^{T}z\left(t_{i-1}\right)+\omega_{d}D\left(t_{i-1}\right)+\omega_{0}\right)\Delta t_{i}\right)}^{2}}{2\sigma^{2}\Delta t_{i}}\right\}. \end{array}$$

### 3.2. Physiological Degradation Process

The changes in some underlying biomarkers associated with AD are monotonic. Two of them are the progressive accumulation of beta-amyloid plaques outside neurons and of tau tangles inside neurons. In addition, brain atrophy (i.e., decreased brain volume) occurs due to cell loss in AD patients, resulting in the loss of cognitive function [26]. In this paper, we generally refer to these changes as the physiological degradation process {D(t),t≥0}, which is considered to be monotonic, i.e., $\Delta D\left(t_{i}\right)=D\left(t_{i}\right)-D\left(t_{i-1}\right)\geq 0$. The gamma process is a commonly used monotonic stochastic process with independent gamma-distributed increments. The physiological degradation process is proposed based on the gamma process and is related to emotional states, which is formulated as follows: (6)$$D\left(t_{i}\right)=D\left(t_{i-1}\right)+Ga\left(\alpha_{D}\left(t_{i},t_{i-1}\right),\beta \right),$$
where the shape parameter $\alpha_{D}\left(t_{i},t_{i-1}\right)$ is strictly greater than zero, defined as follows: (7)$$\alpha_{D}\left(t_{i},t_{i-1}\right)=c_{ga}f_{s}\left(M\left(t_{i-1}\right)\right)\Delta t_{i},$$
where $f_{s}\left(\cdot \right)$ is a sigmoid function (or a logistic function), and $c_{ga}$ is the coefficient. Similar to the tanh(·) function, the sigmoid function is a bounded S-like function, but it has a value range from 0 to 1. During the time period $\left[t_{i-1},t_{i}\right)$, given the emotional state at the last time $M(t_{i-1})$, the probability density function of $\Delta D(t_{i})$ is formulated as follows: (8)$$f_{\Delta D(t_{i})}\left(\Delta d_{i}|M\left(t_{i-1}\right)\right)=\frac{\beta^{c_{ga}f_{s}\left(M\left(t_{i-1}\right)\right)\Delta t_{i}}}{\Gamma (c_{ga}f_{s}\left(M\left(t_{i-1}\right)\right)\Delta t_{i})}\Delta {d_{i}}^{c_{ga}f_{s}\left(M\left(t_{i-1}\right)\right)\Delta t_{i}-1}\exp\left\{-\beta \Delta d_{i}\right\},$$
where Γ(·) is the gamma function, defined as $\Gamma \left(z\right)=\int_{0}^{\infty }x^{z-1}e^{-x}dx$.

### 3.3. Observation Model Based on Physiological Degradation and Emotional State

In clinical practice, the data we can directly obtain from the subjects are called observations, while physiological degradation and emotional states are latent variables and cannot be directly accessed. Cognitive scales are commonly used to assess the severity of AD, which are considered to be accessible observations. As one of the commonly used cognitive scales, MMSE is selected in this paper due to its high sensitivity regarding AD [27]. A decrease in this scale’s results indicates more severe AD. In this subsection, an observation process related to physiological degradation and emotional state is proposed. The observation process is defined based on the linear regression model as follows: (9)$$Y\left(t_{i}\right)=h_{0}+h_{d}D\left(t_{i}\right)+h_{m}M\left(t_{i}\right)+\epsilon ,$$
where the residual ϵ follows normal distribution $N(0,\sigma_{\epsilon })$, and $h_{0}$, $h_{d}$, and $h_{m}$ are regression coefficients. The physiological degradation process in this paper is characterized by a monotonically increasing process, with larger values indicating a more severe condition. The emotional state is represented by a positive or negative continuous value. When it is greater than zero, and the value is larger, it means that the emotional state will accelerate the physiological degradation process, and, when the opposite occurs, it will slow down the degradation process. The observations available in clinical practice may increase or decrease (i.e., in the opposite direction of physiological degradation) as the disease progresses. The sign of the regression coefficients satisfies the mapping in these two cases. Given D(t) and M(t), the probability density function of Y(t) is formulated as: (10)$$f_{Y(t)}\left(y_{t}|D\left(t\right),M\left(t\right)\right)=\frac{1}{\sqrt{2\pi }\sigma_{\epsilon }}\exp\left\{-\frac{{\left(y_{t}-\left(h_{0}+h_{d}D\left(t\right)+h_{m}M\left(t\right)\right)\right)}^{2}}{2{\sigma_{\epsilon }}^{2}}\right\}.$$

As described in Section 3.1, Section 3.2 and Section 3.3, so far we have obtained a complete coupling model of physiological degradation and emotional state for the prediction of AD. From time $t_{0}$ to $t_{n}$, the probability of the entire observation process is obtained by iterating the three formulas, summarized as follows: (11)$$\left\{\begin{array}{c} M\left(t_{i}\right)=M\left(t_{i-1}\right)+\mu_{M}\left(t_{i},D\left(t_{i-1}\right)\right)+\sigma B\left(\Delta t_{i}\right)\\ D\left(t_{i}\right)=D\left(t_{i-1}\right)+Ga\left(\alpha_{D}\left(t_{i},t_{i-1}\right),\beta \right)\\ Y\left(t_{i}\right)=h_{0}+h_{d}D\left(t_{i}\right)+h_{m}M\left(t_{i}\right)+\epsilon \end{array}\right.$$

Due to the nonlinear and non-Gaussian nature of this model, the marginal distribution of the observation process cannot be expressed explicitly. Therefore, Markov chain Monte Carlo (MCMC) methods are utilized to estimate posterior distributions of the model parameters.

## 4. Experiments

### 4.1. Experiment on Synthetic Data

We first conducted an experiment on synthetic data to perform a preliminary validation of the proposed P-E model. This subsection focuses on the effects of emotional impacts and the parameters of the model on the disease progression, thus providing guidance for applying the P-E model to real datasets. We validated the performance of the P-E model in predicting observation sequences and compared the performance with some state-of-the-art prediction models.

#### 4.1.1. Synthetic Data Generation

In the P-E model, only the trajectory of the observed variables can be directly assessed, while the processes of physiological degradation and emotional state transition are not observable. In the synthetic data experiment, in order to correspond to the cognitive test results in the real dataset, we design the value range of the observed variable to be from 0 to 30. A score of 30 indicates no cognitive impairment, and a lower score means more severe cognitive impairment. A total of five groups of observation sequences with different parameters were generated, and each group contained 20 sequence samples from 21 observations. The parameter settings for each group are shown in Table 1. In this experiment, our focus is not to consider the influence of the covariates z(t) on the model, so we set the regression coefficient $\omega_{z}$ to zero. The parameter $c_{ga}$ in Setting 1 and Setting 2 is set differently, in order to illustrate the effect of emotional state on physiological degradation. In Settings 3, 4 and 5, the settings of the parameters $k^{\left(+\right)}$ and $k^{\left(-\right)}$ are different, and we keep the parameters $\lambda^{\left(+\right)}$ and $\lambda^{\left(-\right)}$ the same. The coefficients $\omega_{\delta }=\left[\omega_{\delta }^{\left(+\right)}\;\omega_{\delta }^{\left(-\right)}\right]$ of the emotional impacts on the emotional state transition are the same, which are 0.1 and −0.1, respectively. Settings 3, 4, and 5 correspond to positive-dominant emotional impacts (i.e., there are more positive emotions than negative emotions), negative-dominant emotional impacts, and fluctuating emotional impacts, respectively.

#### 4.1.2. Synthetic Data Analysis

In this paper, MCMC methods were used for estimating the posterior distributions of the parameters. For each setting, we selected 10 samples for parameter estimation, and the remaining samples were used for prediction tasks. We set the number of warm-up iterations to 6000 and obtained Bayesian results from 4000 iterations after the warm-up iterations. The number of chains is set to three, so as to check the convergence. Taking Setting 1 as an example, as shown in Figure 3, the three chains corresponding to each parameter converged around the same value after the warm-up iterations. The posterior probability distributions are shown in Figure 4. The posterior distributions estimated by the three chains are coincident, which also shows the convergence of the parameter estimation.

With the estimated posterior distributions of the parameters, we made predictions on the observation sequences in the testing dataset. In each setting, we used 10 samples for the prediction task. Each sample contains 21 observations, corresponding to observation times $\left\{t_{0},\ldots ,t_{20}\right\}$. The prediction task studied in this paper is to predict the observation at the time $t_{i+1}$ given the observations before time $t_{i}$, that is, to obtain $P\{Y\left(t_{i+1}\right)|Y\left(t_{0}\right),\ldots ,Y\left(t_{i}\right)\}$. Each time a new observation is obtained, the parameters of the model are updated, and then the observation $Y(t_{i+1})$ is predicted. In a medical setting, the time intervals are usually not uniform. Our P-E model is a continuous-time model and has natural adaptability to non-regular time series. Therefore, without a loss of generality, we generated time series with uniform intervals in synthetic data experiments.

Figure 5 shows the prediction results of Settings 1 and 2. The black curves represent the true values of the observations, and the red curves represent the predictive values. The red point at time $t_{i+1}$ is predicted from the black points before time $t_{i}$. The red band represents the 5% to 95% prediction interval. It can be seen that the red prediction interval almost covers the black curve, thus the predicted value matches the true value. The blue curves represent the physiological degradation processes D(t), and the parameters $c_{ga}$ of Settings 1 and 2 are 10 and 5, respectively. The value of $c_{ga}$ in Setting 1 is larger, so it results in a faster monotonically increasing D(t) and, thus, a more significant decrease in the observations.

Figure 6 shows the prediction results of Settings 3, 4 and 5. In these three settings, the effect of emotional impacts is explored. In Setting 3, as shown in Figure 6A, the pink area represents positive emotions, and the blue area represents negative emotions. The pink area is larger than the blue area, which indicates that the positive emotions dominate, resulting in a positive shift in the emotional state (yellow curve). Given the parameter $c_{ga}$, the greater the positive emotional state, the faster the physiological degradation process, resulting in a more significant drop in the observations. In Setting 4, as shown in Figure 6B, the negative emotions dominate, and the yellow curve develops negatively, thereby slowing down the physiological degradation process, and the decline in observations is not significant. In Setting 5, as shown in Figure 6C, the negative emotions dominate between times $t_{0}$ and $t_{10}$, and the positive emotions dominate between $t_{11}$ and $t_{20}$. Therefore, the emotional state initially shifts to the negative and then to the positive. The physiological degradation process shows a gentle upward trend at the beginning, and then it increases rapidly. In the observations, it shows a slight decrease at the beginning and then a significant decrease.

### 4.2. Prediction of Alzheimer’s Disease Progression

#### 4.2.1. Data Preparation

The real data used in the experiments were obtained from the Alzheimer’s Disease Neuroimaging Initiative (ADNI) database (adni.loni.usc.edu, accessed on 11 November 2020). The ADNI was launched in 2003 as a public–private partnership, led by Principal Investigator Michael W. Weiner, MD. The primary goal of ADNI has been to test whether serial magnetic resonance imaging (MRI), positron emission tomography (PET), other biological markers, and clinical and neuropsychological assessments can be combined to measure the progression of mild cognitive impairment (MCI) and early Alzheimer’s disease (AD). We treated subjects diagnosed as CN at all follow-ups as normal aging, which was distinguished from neurodegenerative dementia. In the real data analysis, we considered the subject’s APOE4 gene as a covariate in the model and did not consider the effect of emotional impacts on the model, so we set the parameter $\omega_{\delta }$ to zero. After excluding subjects with normal aging and missing APOE4 gene information, we obtained MMSE scores from 134 subjects with more than 10 follow-ups. We divided the 134 subjects into a training set (106 subjects) and a testing set (28 subjects). The training set was used to estimate the model parameters, and the testing set was used to evaluate the predictive performance of the model.

#### 4.2.2. Alzheimer’s Disease Progression Analysis

In Section 4.1.1, we generated synthetic data based on characteristics of cognitive data in AD. In the AD experiment, we heuristically selected prior distributions of parameters based on the parameters in the synthetic data experiment. We used the MCMC method to estimate the posterior distributions of parameters of the model. The selected prior distributions and estimated posterior distributions of parameters are shown in Table 2. The R-hat convergence diagnostic compares the between- and within-chain estimates for model parameters [28]. If the R-hat is less than 1.05, then the chains have mixed well. In the results of the posterior distribution estimation, the R-hats of all of the parameters were less than 1.05, so we considered the estimation results to be convergent.

With the posterior probability distributions of the parameters, we evaluate the prediction performance on the testing set. As there are regular and random missing data in the follow-ups of each subject, the interval time between two observations of the same subject is not constant. When we evaluated the prediction performance of the model, we used the posterior distributions of the parameters obtained from the training set as the prior distributions of those of the testing set. As new observations of the testing subject were obtained, the posterior distributions of the parameters were updated, and the MMSE score at the next observation time was predicted.

Recurrent neural networks (RNNs) are commonly used in time series with uniform intervals. However, in medical settings, such as the prediction task studied in this paper, they are an awkward fit for irregularly sampled time series data. Therefore, we selected four variants of the RNN to compare with the proposed P-E model, including the continuous-time RNN (CT-RNN) [29], the ordinary differential equation RNN (ODE-RNN) [30], the continuous-time LSTM (CT-LSTM) [31], and gated recurrent units with ODE (GRU-ODE) [32]. We used the mean squared error (MSE) as an evaluation metric for the prediction performance.

We performed five runs of each predictive model on the testing set with different initial settings. The mean and standard deviation were calculated for the MSE of the five runs, and the results are shown in Table 3. Our P-E model achieved an MSE of 7.137±0.035, which is the best among all of the compared methods. In Figure 7, the prediction results of three of the testing subjects are shown as examples. It can be seen that our P-E model can achieve a good prediction performance for subjects with slow or rapid development of AD progression.

## 5. Discussion

The proposed P-E model is a general model that can be applied to all stages of AD. With the development of a subject’s AD progression, new observations are continuously obtained, and we continuously update the parameters of the model to reflect the differences at different stages. Based on the results of the synthetic data experiment and the AD experiment, it can be seen that our P-E model has the ability to make good predictions on the MMSE scores of AD patients. Since we introduce a non-monotonic emotional transition process into the model, the model can also achieve good predictions for observation sequences that are not monotonically decreasing. At the same time, due to consideration of the covariates z(t), the model can also be adapted to prediction tasks with individual differences. Deep learning-based methods, which often require a large amount of data, do not work with a transparent mechanism, so it is difficult to take into account the physiological process in AD, resulting in a predictive performance that is not as good as in the classical application domains.

At present, our P-E model has been validated to characterize the development of AD and make predictions. However, there is still some work to be done in the future. The effect of emotional impacts on AD development has only been validated in synthetic data. When the model is applied to AD, further work is needed to explore how to quantify patients’ experiences in daily life as emotional impacts. In addition, the physiological degradation process is considered to be a hidden process in the current model, which is a general representation of the monotonic physiological indicators of AD patients. In future works, we intend to replace this current form of hidden process with observable physiological degradation processes. In this way, this process in the model can adopt supervised learning, allowing for a clearer physiological interpretation and helping reveal the pathogenesis of AD.

## 6. Conclusions

This paper proposes a novel coupling model of the physiological degradation and emotional state (P-E model) for AD prediction. Through the synthetic data experiment, the basic performance of the model was validated, and the effects of the parameters of the model on the observations were discussed. In the real data experiment, the model was validated to be effective in AD and achieved the best prediction performance compared with other state-of-the-art methods. In addition, the P-E model provides an explanation for the non-monotonicity in AD progression. Thus the high interpretability of this model can help researchers to further understand AD pathogenesis.

## Figures and Tables

**Figure 1 brainsci-12-01132-f001:**
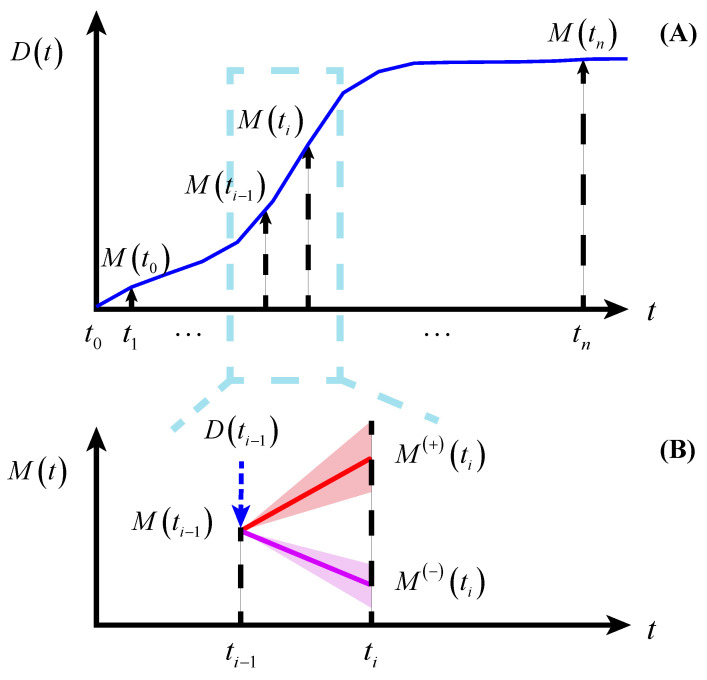
The relationship between physiological degradation and emotional state transition. (**A**) The physiological degradation process. (**B**) The emotional state transition process.

**Figure 2 brainsci-12-01132-f002:**
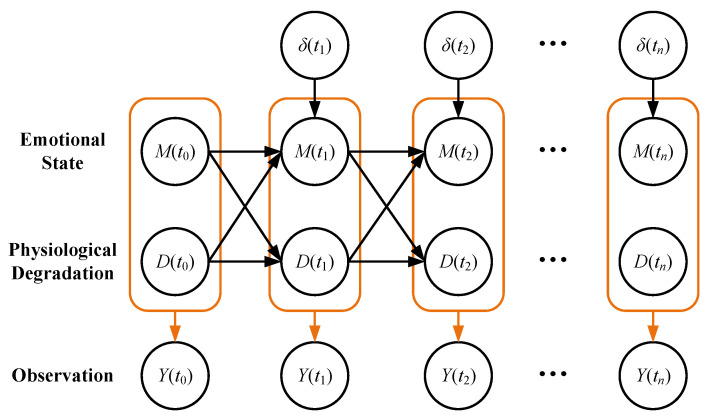
A novel coupling model of physiological degradation and emotional state.

**Figure 3 brainsci-12-01132-f003:**
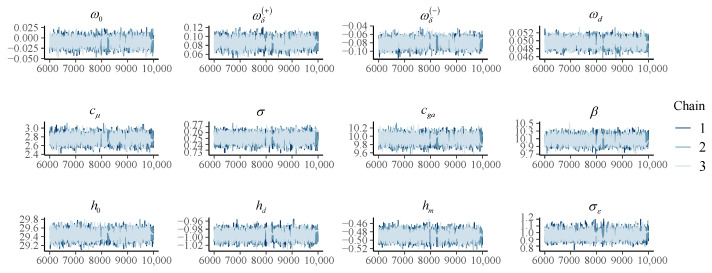
Convergence diagnostics.

**Figure 4 brainsci-12-01132-f004:**
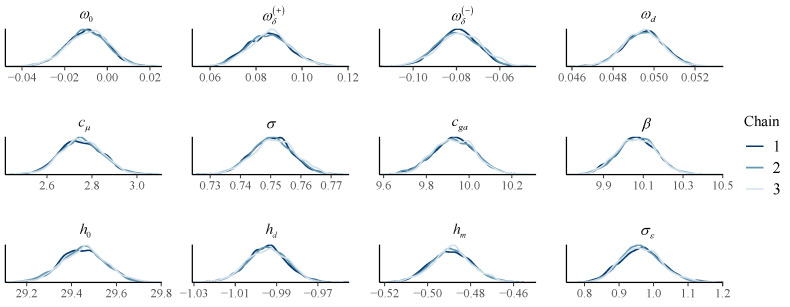
Bayesian parameter estimation.

**Figure 5 brainsci-12-01132-f005:**
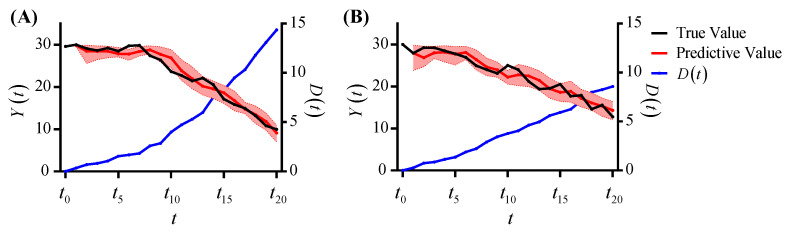
The prediction results of under different $c_{ga}$. (**A**) Setting 1. (**B**) Setting 2.

**Figure 6 brainsci-12-01132-f006:**
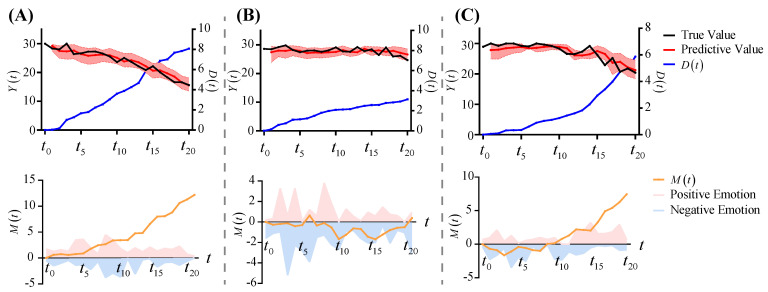
The prediction results under different emotional impacts. (**A**) Setting 3. (**B**) Setting 4. (**C**) Setting 5.

**Figure 7 brainsci-12-01132-f007:**
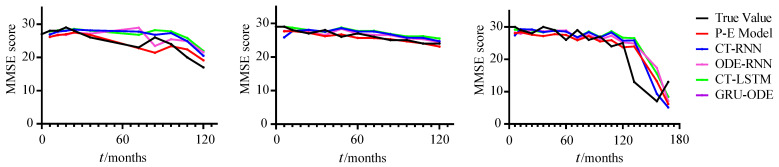
The prediction results of AD.

**Table 1 brainsci-12-01132-t001:** Parameter settings for synthetic data.

	$k^{\left(+\right)}/k^{\left(-\right)}$	$\lambda^{\left(+\right)}/\lambda^{\left(-\right)}$	$\omega_{\delta }$	$\omega_{d}$	$\omega_{0}$	$c_{\mu }$	σ	$c_{\mathrm{ga}}$	β	$h_{0}$	$h_{d}$	$h_{m}$	$\sigma_{\epsilon }$
1	5/4	3/3	0.1/−0.1	0.05	0	3	0.5	10	10	30	−1	−0.5	1
2	5/4	3/3	0.1/−0.1	0.05	0	3	0.5	5	10	30	−1	−0.5	1
3	5/3	3/3	0.1/−0.1	0.05	0	3	0.5	5	10	30	−1	−0.5	1
4	3/5	3/3	0.1/−0.1	0.05	0	3	0.5	5	10	30	−1	−0.5	1
5	3&5	3/3	0.1/−0.1	0.05	0	3	0.5	5	10	30	−1	−0.5	1

**Table 2 brainsci-12-01132-t002:** The selected prior distributions and estimated posterior distributions of parameters.

Prior Distributions			Posterior Distributions			
**Parameters**	**Mean**	**Standard Deviation**	**Parameters**	**Mean**	**Standard Deviation**	**R-Hat**
$\omega_{d}$	0.10	0.05	$\omega_{d}$	0.03	0.01	1.00
$\omega_{1}$	0.30	0.05	$\omega_{1}$	0.16	0.03	1.00
$\omega_{0}$	−0.10	0.05	$\omega_{0}$	−0.16	0.04	1.00
$c_{\mu }$	1.30	0.05	$c_{\mu }$	1.29	0.05	1.00
σ	1.00	0.05	σ	1.11	0.04	1.00
$c_{ga}$	5.00	0.10	$c_{ga}$	4.98	0.10	1.00
β	6.00	0.10	β	6.02	0.10	1.00
$h_{0}$	29.50	0.05	$h_{0}$	29.49	0.05	1.00
$h_{d}$	−0.50	0.05	$h_{d}$	−0.41	0.05	1.04
$h_{m}$	−1.00	0.05	$h_{m}$	−0.86	0.04	1.01
$\sigma_{\epsilon }$	0.80	0.05	$\sigma_{\epsilon }$	1.05	0.03	1.00

**Table 3 brainsci-12-01132-t003:** Prediction of MMSE scores. (mean ± standard deviation, *N* = 5).

Model	MSE
P-E Model (ours)	7.137 ± 0.035
CT-RNN	14.752 ± 1.474
ODE-RNN	13.498 ± 0.278
CT-LSTM	22.369 ± 0.590
GRU-ODE	14.315 ± 1.706

## Data Availability

The original data used in this project and that support the findings of this study are openly available through https://ida.loni.usc.edu (accessed on 11 November 2020) upon approval from the ADNI project administration.

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
