# Peer review of "A Novel Coupling Model of Physiological Degradation and Emotional State for Prediction of Alzheimer’s Disease Progression"

_brainsci, 2022, doi:10.3390/brainsci12091132_

Round 1

Reviewer 1 Report

Here, the authors propose a novel model for predicting the trajectory of cognitive deterioration in Alzheimer cases which couples continuous physiological degradation with stochastic changes in emotional status of the cases (P-E model). To develop the model synthetic data was first used with MMSE as the outcome measure. The physiological degeneration was considered to be a continuous irreversible  process reflected by MRI volume change while increases and decreases in emotional status such as depression and anxiety were stochastic and unpredictable. To validate the model it was then tested on 134 subjects each with over 10 follow-ups from the Alzheimer’s Disease Neuroimaging Initiative (ADNI). The prediction performance of the P-E model was significantly better than other machine learning algorithms  for characterising MMSE decreases on an individual level. The authors conclude that their P-E model provides an explanation for the non-monotonicity in AD progression and can help researchers to further understand AD pathogenesis.

This is a novel report as, while other models of AD progression already exist, they do not take into account the stochastic effects of emotion on patient recovery or deterioration. I have some comments:

First, the ADNI cohort, while large, is not ideal for the validation of this model as the MMSE is a relatively insensitive and non-linear measure of AD progression in early disease and the database does not contain detailed ratings of emotional status at each visit. As a consequence one cannot state that this model has been validated with real life data. There are other smaller public databases which may be more suitable. Alternatively, a longitudinal study needs to be set up that is designed  to validate the model with frequent ratings of emotional status and a more sensitive global cognitive assessment.

Second, while volumetric MRI can detect small changes in volume accurately in AD, these only become evident later in the disease and there are now more sensitive MRI markers of microstructure and connectivity available in public databases which could be entered into this model.

In conclusion while the P-E model could potentially be used to accurately predict the progression of individual AD cases and explain their apparent improvements at times, my feeling is that it has not yet been adequately validated with real-life data.  

Reviewer 2 Report

Authors have done a good approach in suggesting a novel model to predict the AD progression.

Though it is unclear that at what stage of the AD disease this model can be used for the prediction. Is there any criteria or parameters to opt the early-stage or later stage AD? As the physiological and emotional indicators involved in this model are observed in both early and late onset of the disease. But the severity and pathogenesis is different in both the stages. Authors should mention and discuss such information in the discussion section. If the model is consistent for every stage of AD and there will be a similar simulation for every age, gender and ethnicity, then authors should mention this too.
